# Investigating SnO$_x$/Graphene Oxide heterostructure for methane sensing and its application as a tunable light absorber for optoelectronic devices

Manoj Kumar[1], Purnendu Shekhar Pandey[2], M. Sudhakara Reddy[3], Anita Gehlot[4], Santosh Kumar Choudhary[5], Gyanendra Kumar Singh [6], Balkeshwar Singh [7]*

1 MLR Institute of Technology, Hyderabad, India, 2 Department of Electronics and Communication Engineering, G.L. Bajaj Institute of Technology and Management, Greater Noida-, Uttar Pradesh, India, 3 Department of Physics & Electronics, JAIN (Deemed to be University), Bangalore, Karnataka, India, 4 Uttaranchal Institute of Technology, Uttaranchal University, Dehradun, India, 5 VNR Vignana Jyothi Institute of Engineering & Technology, Hyderabad, India, 6 Department of Mechanical Engineering, Greater Noida Institute of Technology, Greater Noida, Uttar Pradesh, India, 7 Department of Mechanical Engineering, Program of Manufacturing Engineering, Adama Science and Technology University, Adama, Ethiopia

* balkeshwar.singh@astu.edu.et

## Abstract

This study investigates the optical and electronic properties of SnO$_x$/Graphene Oxide (SnO$_x$/GO) heterostructures, focusing on their sensitivity and selectivity to methane adsorption and its tunable light absorption capabilities across different wavelength ranges. By categorizing SnO$_x$/GO heterostructures into four types based on the oxygen mole fraction (x) of SnO$_x$, notable differences are observed in their light absorption, extinction coefficient, and reflectance. Among these, *Type-C* heterostructures demonstrate the highest absorption coefficient (~$1.8 \times 10^5$ cm$^{-1}$), indicating strong potential for UV and visible light applications. Building upon the optimized *Type-C* SnO$_x$/GO heterostructure, we further examine the effect of varying concentrations of methane molecules adsorbed on its surface. This leads to the classification of four additional heterostructure types- *Type-I* to *Type-IV* which are based on the methane molecules concentration adsorbed on the surface of an optimized SnO$_x$/GO heterostructure. The interaction with methane further modulates the optoelectronic properties of heterostructure, with *Type-II* heterostructures demonstrating the highest extinction coefficient (~8.0 at 1000 nm) and strong near-infrared absorption. In contrast, *Type-IV* structures, characterized by the highest methane concentration, show a significant increase in reflectance (~0.85) and a reduction in absorption. Additionally, an energy distribution analysis of various atmospheric gases, such as CH$_4$, H$_2$O, and CO$_2$ were conducted to evaluate the selectivity of SnO$_x$/GO heterostructure based sensors. The aim was to ensure minimal interference from other ambient gases. The analysis revealed that CH$_4$ exhibits a more negative energy state, indicating higher

**Data availability statement:** All relevant data are within the paper and its Supporting information files.

**Funding:** The author(s) received no specific funding for this work.

**Competing interests:** The authors have declared that no competing interests exist.

stability and a greater affinity for adsorption on the sensor surface compared to the other atmospheric gases. This stabilization highlights the interaction dynamics of the material, reinforcing its potential for diverse applications, including UV absorption, infrared transparency, and trace methane detection. Overall, these findings establish $SnO_x$/GO heterostructures, particularly the *Type-C* variant with an optimal oxygen mole fraction (x), as promising candidates for advanced optical and methane gas-sensing technologies.

## Introduction

The persistent quest for new advances in gas sensing and optoelectronics has led to significant research in terms of new materials and device architectures. This work explores fascinating properties of a $SnO_x$/GO heterostructure, highlighting its dual potential for highly sensitive and selective methane detection, as well as tunable light absorption for optoelectronic applications [1,2]. It is well known fact that methane ($CH_4$) is a powerful greenhouse gas with a much higher global warming potential than carbon dioxide, making it a serious threat to the environment. Its rising concentration in the atmosphere, mainly caused by human activities, highlights the need for reliable and sensitive detection methods for environmental monitoring, industrial safety, and climate change efforts [3,4]. Traditional methods of methane detection often suffer from disadvantages such as high-power consumption, bulky instrumentation, and limited sensitivity, which makes highly sensitive, selective, and energy-efficient methane sensors important [5].

There are several materials available for methane detection, but metal oxides stand out due to their low operating temperatures (ranges around 25°C – 250 °C), high sensitivity, and stability [6]. One of the key advantages of metal oxides is their ease of fabrication at low temperatures using well-established techniques such as sputtering, sol-gel, and metal-organic chemical vapor deposition (MOCVD), making them an ideal choice for sensor applications. In our study, we have selected metal oxide ($SnO_x$) for methane sensing in combination with graphene oxide, leading to the formation of the $SnO_x$/GO heterostructure. This heterostructure offers several advantages, including enhanced sensitivity, improved selectivity, reduced operating temperatures for methane detection, and tunable optoelectronic properties [7]. In comparison to pure $SnO_2$ or graphene-based sensors, the $SnO_x$/GO heterostructure offers several distinct advantages. The integration of $SnO_2$ with graphene oxide (GO) significantly enhances charge transfer and surface interactions, improving the overall sensitivity and selectivity of the sensor [2]. The heterostructure benefits from the high surface area of GO, which facilitates better adsorption of target molecules, while the semiconducting properties of $SnO_2$ provide a robust platform for gas detection. Additionally, the hybrid structure offers better stability, improved conductivity, and increased photocatalytic activity, which are advantageous for applications in methane sensing [8]. These combined properties make the $SnO_x$/GO heterostructure more more efficient in methane detection, compared to pure $SnO_2$ or graphene

based sensors, which may not exhibit the same level of sensitivity and stability. The $SnO_x$, a well-known metal oxide, is renowned for its tunable bandgap and high stability, making it ideal for gas sensing and optoelectronic applications [9]. GO, on the other hand, is a two-dimensional material with remarkable surface area and tunable electronic properties [10], which enhances the performance of the composite when integrated with $SnO_x$. The combination of the optical and electronic properties of graphene oxide (GO) with the semiconducting nature of $SnO_x$-based composites establishes a robust and versatile platform for the development of optoelectronic devices. While this study is centered on theoretical investigations, it builds upon a strong foundation of experimentally validated synthesis methods for $SnO_x$/GO-based heterostructures, which are well-documented in existing literature. According to literature, $SnO_x$/GO nanocomposites have been effectively synthesized using straightforward and cost-efficient methods, including room-temperature solution-based approaches and hydrothermal techniques [2,11]. A single-step method employing a domestic pressure cooker has also been reported for the fabrication of reduced graphene oxide/$SnO_2$ (RGO/$SnO_2$) composites, demonstrating the feasibility of scalable synthesis [12] Furthermore, a combination of sol-gel and hydrothermal methods has been used to produce $SnO_2$/rGO composites with tunable dielectric properties, directly relevant to sensor applications [13].In terms of material stability, experimental evidence indicates that $SnO_2$/graphene-based sensors maintain excellent methane sensing performance and structural integrity over extended testing periods [14]. Regarding its stability, multiple studies have consistently demonstrated the robust and durable performance of $SnO_x$/GO-based materials when exposed to methane under ambient conditions. In a study, it is reported that GO-$SnO_2$ composites exhibits stable methane sensing responses without significant degradation, even after repeated testing and prolonged ambient storage [15]. Supporting this, an earlier study confirmed the long-term operational reliability of related nanocomposite systems, emphasizing their durability [16]. Furthermore, additional evidence proves that Pd-doped $SnO_2$/rGO nanocomposites maintained consistent sensing behavior over long periods at room temperature [17]. Collectively, these findings affirm the suitability of $SnO_x$/GO-based sensors for reliable and sustained methane detection in practical applications. Ultimately, this research aims to pave the way for the next generation of sensors and devices, offering improved performance and functionality by uncovering the core mechanisms behind the gas sensing and optoelectronic capabilities of $SnO_x$-GO heterostructure.

## Design consideration

In this study, the CASTEP toolkit package of Material Studio (BIOVIA) was utilized to explore the tunable optical and electronic properties of the $SnO_x$/GO heterostructure. Additionally, the Adsorption Locator tool in Materials Studio was employed to analyze the energy distribution of various atmospheric gases, including $CO_2$, $H_2O$, and $CH_4$, adsorbed on the surface of the $SnO_x$/GO heterostructure. It may be mentioned here that the dimensions of the simulation cells for each configuration of $SnO_x$/GO heterostructure has an in-plane unit cell of approximately 15 Å × 15 Å with a vacuum layer of 20 Å along the z-axis. The calculations have been performed using generalized gradient approximation (GGA) with the Perdew-Burke-Ernzerhof (PBE) functional by norm-conserving pseudopotentials of Cambridge Sequential Total Energy Package code (CASTEP) tool kit of Material Studio. Specifically, norm-conserving pseudopotentials has been used for all elements and incorporated Grimme's correction for van der Waals interactions in all calculations to account for long-range dispersion forces. Regarding k-point sampling, we set the grid to 2×2×2 for Brillouin zone integration in bulk systems, ensuring sufficient accuracy for the simulations while maintaining computational efficiency. The convergence tolerance parameters value was set to be as 830 eV as cutoff energy for the k-point with fine mesh, maximum force, maximum stress and maximum displacement were set to be 0.03 eV/$A^0$, 0.05 GPa and 0.001 Å, respectively. The SCF tolerance value was considered to be fine, i.e., $10^{-9}$ eV/atom.

Fig 1 illustrates the molecular structure of $SnO_x$/GO with varying oxygen content (x) in $SnO_x$. The oxygen molecule content in $SnO_x$ was systematically varied, and its corresponding optoelectronic properties were observed. In Fig 1a–d, Sn atoms are depicted in green, oxygen atoms in pink, and the graphene oxide layer, positioned over the $SnO_x$ layer, is shown in gray. The bottom $SnO_x$ layer exhibits a gradual decrease in oxygen mole fraction which is shown from Fig 1a–d.

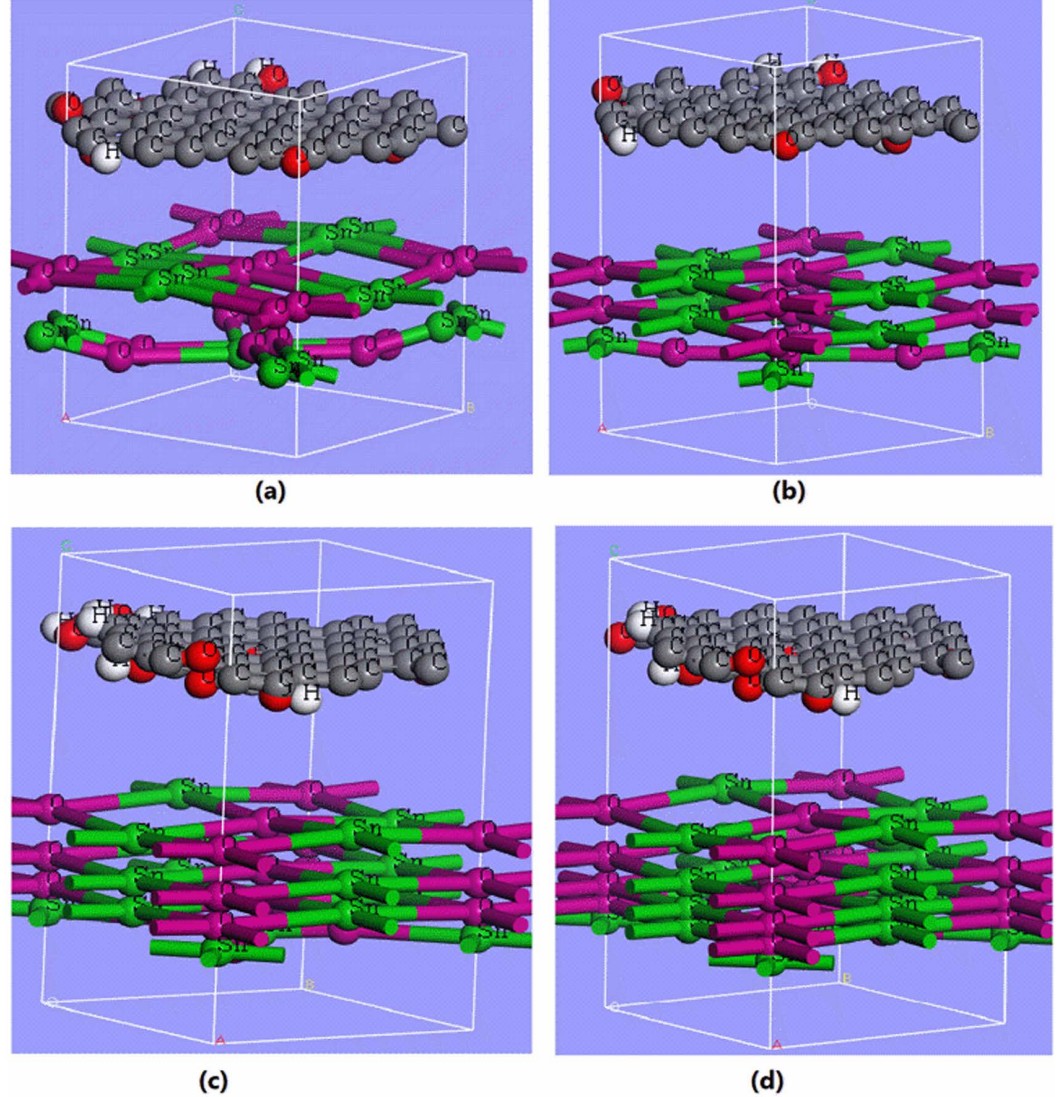

**Fig 1. Representation of the SnO$_x$/GO heterostructure, showcasing varying oxygen mole fraction (x) in SnO$_x$ layer while preserving the integrity of the graphene oxide layer.**

Fig 1a represents a relatively higher oxygen mole fraction in SnO$_x$, while Fig 1d illustrates the lowest oxygen mole fraction. Throughout these molecular arrangements, the top graphene oxide layer remains unchanged. We have performed Density Functional Theory (DFT) simulations to model SnO$_x$ molecules with varying oxygen content (x). The non-stoichiometry of SnO$_x$ was systematically adjusted by modifying the relative proportions of SnO and SnO$_2$ [18,19].

For each compositional ratio, we calculated the corresponding value of (x), representing the oxygen mole fraction or the average number of oxygen atoms per Sn atom. In this study, we have considered four different ratios of SnO$_2$ and SnO to systematically vary the oxygen content in SnO$_x$, resulting in compositions with (x)=1.5, (x)=1.33, (x)=1.25, and (x)=1.20. These non-stoichiometric compositions represent intermediate states between pure SnO$_2$ and SnO are categorized as follows: SnO$_{1.5}$ (*Type-A*, oxygen-rich), SnO$_{1.33}$ (*Type-B*), SnO$_{1.25}$ (*Type-C*), and SnO$_{1.20}$ (*Type-D*, oxygen-deficient). The SnO$_{1.5}$ corresponds to a mixture of 1 mole of SnO and 1 mole of SnO$_2$, yielding an average oxygen-to-tin ratio (x)=1.5. The value

of oxygen mole fraction (x) is calculated using the formula, (x) = (Number of O atoms in the composition/ Number of Sn atoms in the composition).

In this case, the total O atoms = 1 (from SnO) + 2 (from $SnO_2$) = 3 and the total Sn atoms = 1 (from SnO) + 1 (from $SnO_2$) = 2, yielding (x)=3/2 = 1.5. Similarly, a composition of $SnO_{1.33}$ corresponds to a mixture of 1 mole of $SnO_2$ and 2 moles of SnO, yielding an average oxygen-to-tin ratio (x)=1.33. For $SnO_{1.25}$, the mixture consists of 1 mole of $SnO_2$ and 3 moles of SnO, resulting in (x)=1.25. Likewise, $SnO_{1.20}$ represents a composition with 1 mole of $SnO_2$ and 4 moles of SnO, giving an average value of (x)=1.20. Each of these compositions reflects a different ratio of $SnO_2$ to SnO and is used to model non-stoichiometric $SnO_x$ phases by varying the oxygen content systematically.

Based upon an optimized $SnO_x$/GO heterostructure in terms of opto-electronic properties, we further examine the effect of varying concentrations of methane molecules adsorbed on its surface. This analysis enables the classification of four additional heterostructure types as shown in Fig 2a–d, i.e., Type-I to Type-IV, which are based on the adsorption of methane molecules (depicted in light green atop heterostructure) on the surface of an optimized $SnO_x$/GO heterostructure. Here, Type-I (Fig 2a) represents the lowest concentration of methane molecules adsorbed on the $SnO_x$/GO surface, and Type-IV (Fig 2d) corresponds to the highest concentration. The concentration of methane molecules was varied to study the resulting changes in optoelectronic properties of $SnO_x$/GO heterostructure. The methane molecule concentration atop the $SnO_x$/GO heterostructure was gradually increased, as illustrated in Fig 2a through Fig 2d. It may be mentioned here that Adsorption Locator tool that uses Monte Carlo calculation has been used to identify possible adsorption configurations of $SnO_x$/GO heterostructure. The Monte Carlo calculation parameters were set to standard ambient conditions, with a temperature of 298 K and a pressure of 1 atm. The simulation has been carried out using Universal Force Field with a cutoff radius of 3 Å applied for van der Waals interactions. Additionally, the number of simulation steps was set to 100,000, ensuring sufficient sampling for equilibrium and reliable results. These parameters were selected to provide a realistic assessment of gas–surface interactions [20].

## Results and discussion

First of all, the absorption coefficient of $SnO_x$/GO heterostructures as shown in Fig 1a–d were studied to understand their light-absorbing capabilities. It is worth noting that $SnO_x$/GO structures with comparatively higher oxygen mole fraction (x) are classified as Type-A $SnO_x$/GO heterostructures. In contrast, Type-B $SnO_x$/GO structures denote lower oxygen mole fractions than Type-A, Type-C structures have even lower fractions than Type-C, and finally, Type-D $SnO_x$/GO represents the lowest oxygen molecule fraction in $SnO_x$. Accordingly, the $SnO_x$/GO heterostructures have been categorized into four types, based on oxygen content in decreasing order. All of these four samples showed a sharp absorption peak close to 100 nm with characteristics of materials known for strong absorption in UV or visible light. Beyond this, however, they showed differences in absorption characteristics in the spectrum range as shown in Fig 3a–d, which indicated material properties or compositions were different. Strong UV and visible absorption may just mean that the materials are useful for blocking or absorbing UV illumination while weak infrared absorption would simply mean an appropriate level of transparency over the infrared region.

It has been observed that the absorption coefficient of the $SnO_x$/GO heterostructure increases with the oxygen content in $SnO_x$, progressing from Type-A to Type-C. However, in Type-D, which has the highest oxygen molecule fraction, the absorption coefficient decreases abruptly. Among all types, the Type-C $SnO_x$/GO heterostructure exhibits the highest absorption coefficient.

These computational results align with experimentally reported trends in the light-absorbing capabilities of $SnO_x$, where the absorption coefficient decreases at very high oxygen molecule fractions [21]. This theoretical analysis confirms that the Type-C $SnO_x$/GO heterostructure, with an intermediate oxygen content (lower than Type-D but higher than Type-A and Type-B $SnO_x$/GO heterostructure), demonstrates the highest absorption coefficient.

Furthermore, our data demonstrate that the adsorption of methane molecules on the surface significantly influences the optical and electronic properties of the $SnO_x$/GO heterostructure. As the concentration of methane on the surface of

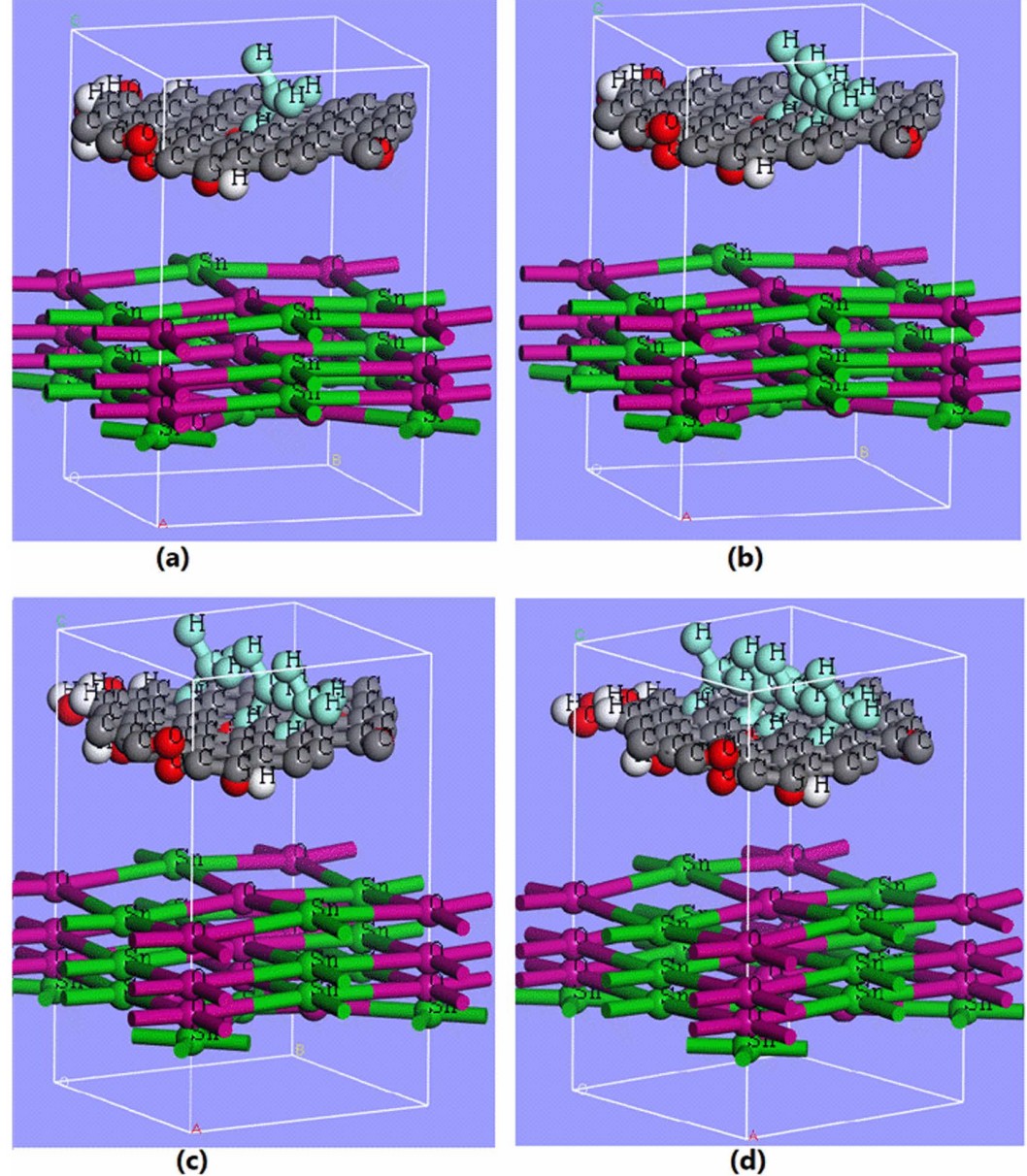

**Fig 2. Illustration of the SnO$_x$/GO molecular structure, highlighting variation of methane concentration atop the heterostructure.**

the SnO$_x$/GO heterostructure increases, notable changes are observed in the refractive index, extinction coefficient, and absorption coefficient. These changes result from the interaction between methane molecules and the SnO$_x$/GO surface, which alters the electronic properties of the SnO$_x$/GO structure. The SnO$_x$/GO heterostructure with varying methane concentrations exhibits wavelength-dependent absorption spectra and reflectivity across the spectral range of 100 nm to 3000 nm, as shown in the Fig 4. Each curve uniquely corresponds to a specific methane concentration adsorbed on the SnO$_x$/GO heterostructure.

Among these, *Type-II* exhibits the highest extinction coefficient, peaking above 8 around 1000 nm, with intense light absorption in the near-infrared region, particularly at longer wavelengths. Conversely, *Type-IV* shows the lowest light

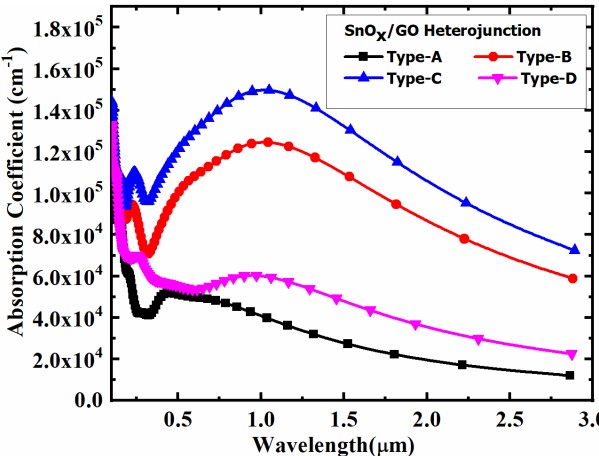

**Fig 3. Absorption coefficient of SnO$_x$/GO heterostructure with variation of oxygen mole fraction in SnO$_x$.**

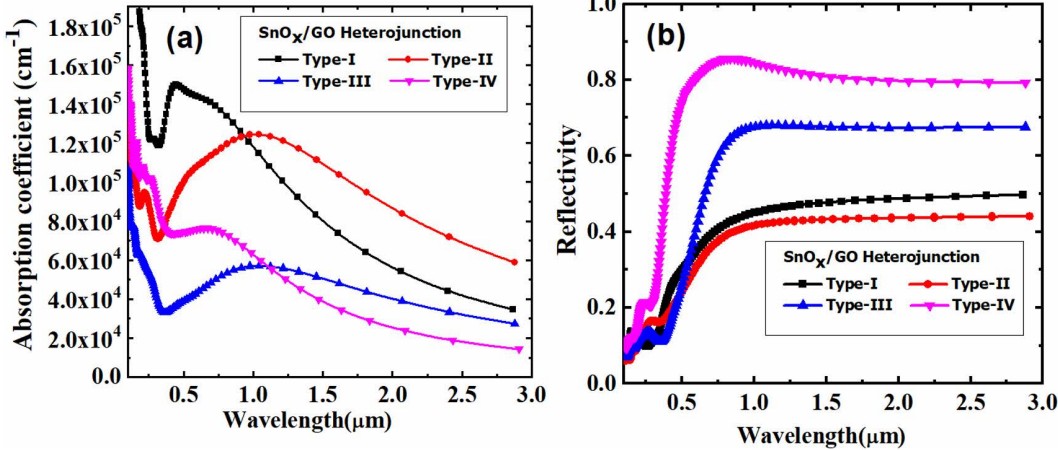

**Fig 4. Absorption coefficient and reflectivity of SnO$_x$/GO heterostructures exposed to different methane concentrations.**

absorption across the entire spectrum. *Type-I* and *Type-III* display moderate absorption, with peaks at approximately 1200 nm and 1500 nm, respectively. Furthermore, the absorption coefficient varies as a function of wavelength, with higher absorption occurring at shorter wavelengths and a gradual decrease at longer wavelengths. Type-I has the highest initial absorption, peaking near $1.6 \times 10^5$ cm$^{-1}$ around 400 nm before gradually declining and flattening in the infrared region beyond 1500 nm. *Type-II* exhibits a minimum around $1.4 \times 10^5$ cm$^{-1}$ and maintains a high absorption coefficient even at longer wavelengths, as it has the least reflectance among all types when measured at longer wavelengths, as seen in Fig 4. *Type-III* and *Type-IV* show significantly different absorption patterns. *Type-III* demonstrates a substantial reduction in absorption compared to other types, falling to negligible levels beyond 1000 nm. *Type-IV*, with the highest methane concentration, exhibits the lowest overall absorption, peaking slightly above $8.0 \times 10^4$ cm$^{-1}$ near 400 nm and steadily decreasing, with minimal absorption in the infrared region beyond 1500 nm.

However, the magnitude of reflectance for SnO$_x$/GO heterostructures with different adsorption concentration of methane, varies significantly among the samples (shown in Fig 4b). Type-IV shows the highest reflectance, peaking at around

0.85. In contrast, Type-I demonstrates a relatively moderate reflectance value of approximately 0.45. Type-II and Type-III fall in between, peaking at around 0.35 and 0.6, respectively. These variations in reflectance highlight specific characteristics of the materials or surface treatments, which are critical for applications in optics, material science, and remote sensing. At lower methane concentrations, reflectance is relatively minimal, accompanied by lower light absorption, as observed in the $SnO_x$/GrapheneOxide heterostructures [22]. Conversely, as the methane concentration increases, reflectance rises within the heterostructure, indicating a lower extinction coefficient. The phenomenon of increased reflectivity and decreased photo absorption caused by methane adsorption on the surface of graphene oxide can be primarily attributed to changes in charge density, density of states (DOS), and the dielectric constant. When methane molecules adsorb onto the graphene oxide surface, they induce a redistribution of electronic charge at the interface, altering the local charge density. This redistribution modifies the electronic environment, leading to subtle shifts in the DOS, particularly near the Fermi level, which affects the probability of electronic transitions under incident light [23]. As shown in the Density of states (DOS) plots in Fig 5, an increase in methane concentration (from Type-I to Type-IV) results in significant modifications in the electronic structure of the $SnO_x$/GO heterostructure.

Specifically, with higher $CH_4$ adsorption, the density of states near the Fermi level becomes more pronounced, indicating stronger electronic interactions between methane molecules and the sensing surface. These changes can be attributed to enhanced charge transfer between methane and the $SnO_x$/GO system. At low concentrations, methane introduces only slight perturbations to the DOS, mostly due to weak physisorption. However, as concentration increases, stronger chemisorption occurs, leading to additional localized states within the bandgap and near the Fermi level. This results in a shift in the Fermi level results in an increased in charge redistribution and electronic coupling between adsorbed methane and surface atoms. These DOS changes directly impact the material's photoabsorption and sensing behavior. The introduction of new states modifies the allowed optical transitions, increasing the absorption of visible and UV light. These findings are consistent with reported trends in literature, where increased DOS near the Fermi level upon gas adsorption correlates with improved sensing properties [24,25]. Furthermore, these electronic changes influence the material's complex dielectric constant—specifically, the real part ($\varepsilon_1$) which governs reflectivity, and the imaginary part ($\varepsilon_2$), which determines absorption. Methane adsorption tends to reduce $\varepsilon_2$ resulting in lower optical absorption, while changes in $\varepsilon_1$ can enhance reflectivity [26]. These combined electronic and optical modifications explain the observed increase in reflectivity and decrease in absorption upon methane adsorption, without requiring any significant structural transformation of the graphene oxide substrate. There is possibility that the high absorption coefficient indeed raises concerns regarding thermal stability and durability [27]. High absorption may lead to an increase in the local temperature during operation, which could negatively impact the material's stability, especially at elevated temperatures. Thermal degradation or material fatigue over prolonged use is a critical factor that must be thoroughly evaluated to ensure the material maintains its performance over time, particularly in environments with fluctuating temperatures [28].

Additionally, while scalability is a key aspect for the practical application of any novel material, fabrication challenges for $SnO_x$/GO heterostructures could arise during the scaling-up process. Achieving consistent quality and uniformity of the heterojunction in large-scale production may require careful optimization of synthesis methods and conditions. Material processing techniques, such as controlled deposition or layer-by-layer assembly, need to be evaluated for their ability to produce $SnO_x$/GO heterostructures on a larger scale without compromising performance [29].

Furthermore, the distinct optical properties exhibited by different $SnO_x$/GO heterostructure types, specifically, the high absorption coefficient of Type-C (~$1.8 \times 10^5$ $cm^{-1}$ at ~100 nm) and the high extinction coefficient of Type-II (~8.0 near 1000 nm), highlight the critical role of oxygen content in tuning the material's optoelectronic response. These two features correspond to different spectral regions and mechanisms of interaction: the absorption coefficient signify how strongly the material absorbs photons (useful for UV or high-energy applications), whereas the extinction coefficient encapsulates both absorption and scattering effects, more relevant for visible to near-infrared applications such as photodetection and gas sensing.

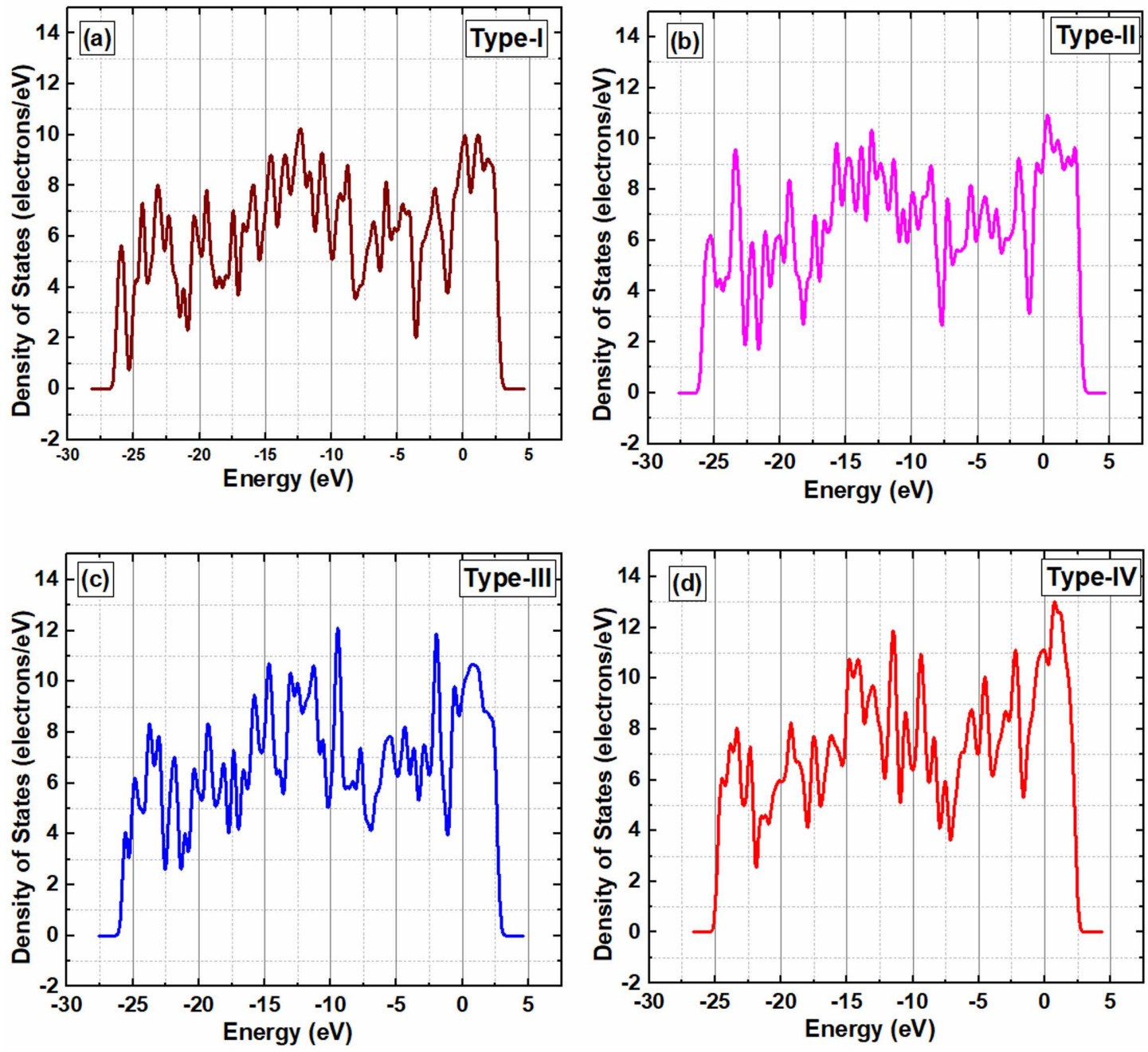

**Fig 5. Density of states (DOS) of the SnO$_x$/GO heterostructure with varying concentrations of methane molecules adsorbed on its surface.**

The potential to synergistically optimize these properties lies in the tunable composition of SnO$_x$. As SnO$_x$ is a mixed-phase oxide comprising varying ratios of SnO (narrower bandgap) and SnO$_2$ (wider bandgap), adjusting the oxygen molar fraction effectively alters the electronic band structure, density of states, and optical transitions. For instance, Type-C SnO$_x$/GO having SnO-rich compositions (relatively having lower bandgap), tend to show enhanced absorption (~1.8 × 10$^5$ cm$^{-1}$ at ~100 nm) in the UV region. This is primarily due to lower energy electronic transitions and stronger localized

electron interactions. In contrast, the Type-II SnO$_x$/GO heterostructure, when exposed to a relatively higher concentration of methane molecules, exhibits significant charge density redistribution and modifications in the density of states. These changes enhance charge separation and dielectric behavior, resulting in a pronounced increase in extinction in the near-infrared (NIR) region. Additionally, methane adsorption plays a significant role in altering the charge distribution, local electronic states, and dielectric properties at the interface of the heterostructure. According to our DFT simulations, the adsorption of methane leads to shifts in the Fermi level, changes in the local charge density, and slight modifications in the optical response due to perturbations in the electronic structure. This suggests that by controlling the exposure to methane, the optical behavior of the material could be dynamically tuned, offering a secondary mechanism for adjusting its properties in response to varying environmental conditions.

In addition, Fig 6 illustrates the wavelength dependent refractive index (n) and extinction coefficient (k), the SnO$_x$/GO heterostructure is highly sensitive to variations in methane concentration, allowing the detection of methane concentrations even below the ppm range. Trace amounts of methane can be effectively determined by analyzing and tracking these coefficients, making this technique highly effective for low-concentration detection. The graph illustrates the reflectance of the four types of structures, Type-I, Type-II, Type-III, and Type-IV, corresponding to one, two, three, and four methane molecules absorbed on the surface of SnO$_x$/GO heterostructure, respectively, as a function of wavelength ranging from 200 nm to 3000 nm. Four distinct samples—Type-I through Type-IV exhibit unique trends for the refractive index and extinction coefficient across the 100 nm to 3000 nm wavelength range.

The refractive index of the SnO$_x$/Graphene Oxide heterostructures varies significantly across the four types. Type-I shows a gradual increase in refractive index, starting at approximately 2 in the 200–300 nm range and plateauing near 4 beyond 2000 nm. Type-II exhibits the most pronounced change, with a dramatic rebound from around 2 at shorter wavelengths to a peak exceeding 8 at the longer wavelength end, indicating a strong refractive response in the near-infrared region. Type-III follows a trend similar to Type-I but with a sharper rise, reaching a peak of about 6 at 2000 nm and continuing to increase thereafter. In contrast, Type-IV displays a smooth increase from about 2 at shorter wavelengths to approximately 5, with much less variation compared to Type-II and Type-III.

Finally, Fig 7 illustrates the energy distribution of various atmospheric gases, including methane (CH$_4$), carbon dioxide (CO$_2$), and water (H$_2$O), on the surface of the SnO$_x$/GO heterostructure. The horizontal axis represents the adsorption energy (E_ads) in kcal/mol, while the vertical axis indicates the probability density of energy, P(E), describing the likelihood of the system having a specific energy [30].

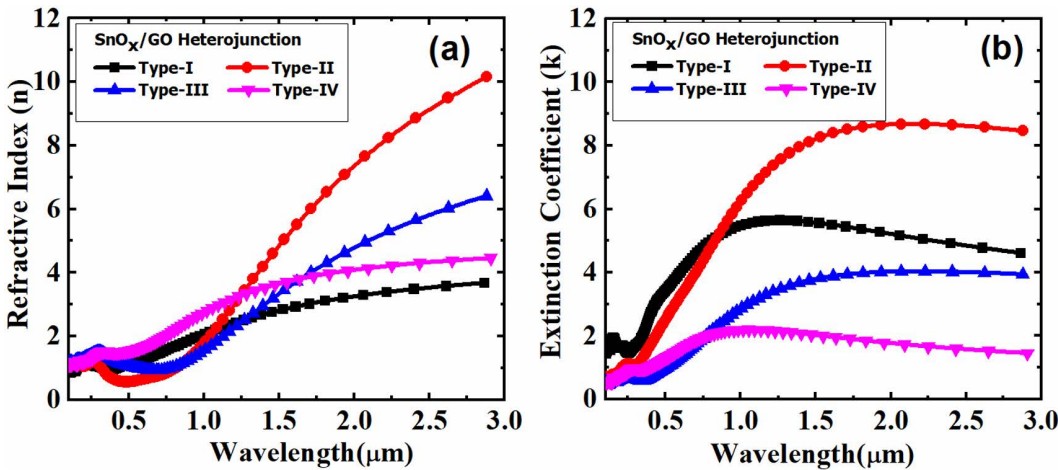

**Fig 6. Wavelength dependent refractive and extinction coefficient of SnO$_x$/GO heterostructure exposed to different methane concentrations.**

Specifically, we have evaluated the adsorption energies of methane ($CH_4$), carbon dioxide ($CO_2$), and water ($H_2O$) on the $SnO_x$/GO surface. Our simulation results indicate that methane exhibits a significantly higher adsorption energy of approximately −78.35 kcal/mol, compared to −42.75 kcal/mol for $H_2O$ and −39.65 kcal/mol for $CO_2$. These results suggest that $CH_4$ interacts more strongly with the sensing surface than the potential interfering gases, supporting the material's preferential affinity for methane. Therefore, under typical environmental conditions, the likelihood of interference from ambient $CO_2$ and $H_2O$ is minimal. This computational validation reinforces the sensor's selectivity and supports its suitability for practical methane detection applications. Moreover, these energy distributions provide important insight into the stability and effectiveness of the sensor. The discrete energy states correspond to different adsorption sites or surface configurations of the $SnO_x$/GO heterojunction [31]. Specifically, a more negative adsorption energy (such as −78.35 kcal/mol) indicates stronger affinity of methane molecules to the surface of $SnO_x$/GO heterostructure, which is typically associated with higher sensing sensitivity and better charge transfer efficiency [32]. This enhanced interaction can influence the local electronic structure, leading to measurable changes in optical properties such as the refractive index and extinction coefficient are key indicators in optical methane sensing. The energy states observed in the system (−78.35, −42.75, and −39.65 kcal/mol) are indeed crucial for understanding the adsorption stability and charge transfer mechanisms of the $SnO_x$/GO heterojunction in methane sensing or photoabsorption processes. Charge transfer between methane molecules and the surface could lead to changes in the electronic properties of the material, impacting its photoabsorption and methane sensing capabilities [33].

The lower energy state (such as −78.35 kcal/mol) typically corresponds to a more stable adsorption configuration, indicating stronger interactions between methane and the $SnO_x$/GO surface. This stronger interaction is likely due to enhanced adsorption strength, which can influence the methane sensing performance by improving the sensitivity of the material to trace amounts of methane [34]. Stronger adsorption generally implies that the material can detect and adsorb methane molecules more effectively, leading to better sensitivity and response. Therefore, this lower energy state could enhance the material's ability to detect even low concentrations of methane, potentially lowering the limit of detection (LOD) and improving overall sensor performance [35].

In summary, $SnO_x$/GO heterojunctions represent a promising class of materials for photoelectrochemical applications, particularly in solar energy conversion systems such as water splitting and $CO_2$ reduction. The heterostructure take

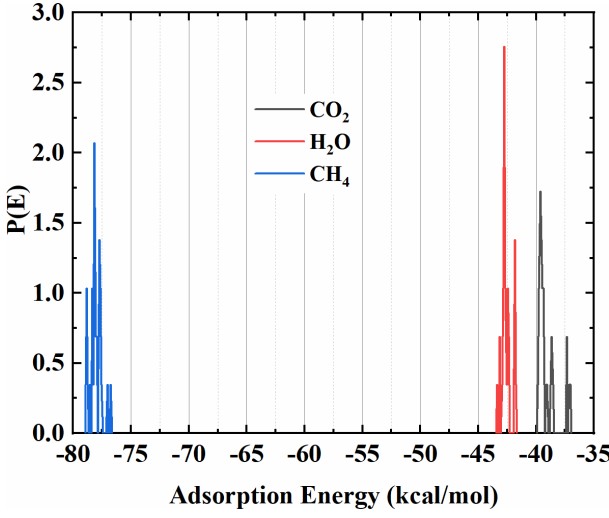

**Fig 7. Energy distribution of $SnO_x$/GO heterostructure for various molecules including carbon dioxide, water and methane, adsorbed atop the heterostructure.**

advantage of its high surface area and electronic properties of graphene oxide (GO), combined with the semiconducting characteristics of tin oxide ($SnO_x$), making them highly efficient in converting light energy into chemical energy. The integration of these two components creates a material with excellent charge separation, enhanced conductivity, and increased photocatalytic activity, making it suitable for use as photoanodes or catalysts in photoelectrochemical cells.

While this study focuses on theoretical investigations, existing experimental research provides a solid foundation that aligns with our theoretical findings. Several studies in the literature have demonstrated practical implementation strategies for $SnO_2$/GO heterojunctions in photoelectrochemical (PEC) applications. These experimental works offer valuable insights into synthesis techniques, device fabrication, and performance characteristics, supporting the potential integration of $SnO_2$/GO heterostructures into real-world PEC systems. For instance, earlier studies have shown that hydrothermal processing is a widely used method for synthesizing $SnO_2$/GO nanocomposites, which significantly enhance photocatalytic and electronic performance. Also, it has been shown that $SnO_2$/GO composites synthesized via this method exhibited improved photocatalytic activity due to enhanced light absorption and charge separation efficiency [2].

Additionally, the sol-gel method has been employed for precise morphological control, which is crucial for tuning the heterojunction's properties. This approach has been used to fabricate $SnO_2$/GO heterostructure which are capable of visible-light-driven photocatalytic degradation, highlighting their potential in environmental remediation and solar-driven chemical reactions. Similarly, chemical vapor deposition (CVD) has been successfully used to grow high-quality $SnO_2$ thin films on GO substrates for optoelectronic applications, demonstrating the scalability and compatibility of CVD for device fabrication [2].

In terms of device integration, a self-powered UV photodetector based on $SnO_2$ nanorods combined with Ag/reduced graphene oxide (rGO) has been developed, showcasing the potential of $SnO_x$/GO-based structures in advanced optoelectronic devices [36]. Furthermore, it has been demonstrated that rGO-$SnO_2$ nanocomposites exhibited strong photocatalytic degradation of methylene blue under both UV and natural sunlight, suggesting their application in waste water treatment and other environmental technologies [37]. Moreover, there is often a significant gap between theoretical simulation results and actual sensing performance. Theoretical studies, such as those based on Density Functional Theory (DFT), are typically conducted under idealized and highly controlled conditions, assuming perfect crystalline structures, isolated gas molecules, and a vacuum environment. These simulations offer valuable insights into fundamental interactions, such as adsorption energies and charge transfer mechanisms. However, simulations often neglect dynamic effects like adsorption/desorption kinetics, response and recovery times, and long-term material degradation. Even if a material demonstrates strong gas interaction theoretically, practical performance may be hindered by challenges in device fabrication, signal transduction, and reproducibility. Therefore, while theoretical models are essential for guiding material design and predicting potential sensing behavior, experimental validation is crucial to confirm these predictions and assess the material's reliability and efficiency in practical applications.

## Conclusion

This study offers a comprehensive insight into the optical and electronic properties of $SnO_x$/GO heterostructures, highlighting its tunable light-absorbing capabilities along with its sensitivity and selectivity towards methane. The classification of $SnO_x$/GO heterostructures into four types based on oxygen mole fraction revealed distinct light absorption characteristics. Among these, *Type-C* heterostructures with intermediate oxygen content demonstrated the highest absorption coefficient, peaking at approximately $1.8 \times 10^5$ cm$^{-1}$ near 100 nm wavelength. This observation aligns with experimental values reported in literature and reinforces the suitability of *Type-C* heterostructures for UV and visible light absorption. Furthermore, methane adsorption on the surface of $SnO_x$/GO heterostructure was investigated, revealing enhanced tunability through variations in the refractive index, extinction coefficient, and absorption spectra. Notably, *Type-II* heterostructures demonstrate the highest extinction coefficient, exceeding 8.0 near 1000 nm, along with strong near-infrared absorption. Additionally, *Type-IV* structures, with the highest methane concentration, show the lowest absorption, peaking at $8.0 \times 10^4$

$cm^{-1}$ near 400 nm, and the highest reflectance (0.85), suggesting saturation effects at elevated gas concentrations. Additionally, energy distribution analysis of various atmospheric gases, such as $CH_4$, $H_2O$, and $CO_2$ on the surface of $SnO_x$/GO heterostructure, reveals that methane ($CH_4$) possesses the most negative energy state, indicating greater stability and a stronger adsorption affinity on the sensor surface. This makes the proposed heterostructure particularly effective for selective methane detection in the presence of other atmospheric gases. Energy distribution analysis reveals that the system stabilizes in discrete energy states, with the most probable state at −8.3 kcal/mol, followed by secondary states at −8.8 kcal/mol and −9.4 kcal/mol. This stabilization underscores the material's interaction dynamics, contributing to its versatility in applications such as UV absorption, infrared transparency, and trace methane detection. These findings highlight the potential of $SnO_x$/GO heterostructures, particularly the *Type-C* variant with optimum amount of oxygen mole fraction, as promising candidates for advanced optical and sensing technologies.

## Supporting information

**S1 File. Data.**
(RAR)

## Author contributions

**Conceptualization:** Manoj Kumar, Santosh Kumar Choudhary.

**Data curation:** Santosh Kumar Choudhary.

**Formal analysis:** Purnendu Shekhar Pandey.

**Investigation:** M. Sudhakara Reddy.

**Methodology:** Anita Gehlot.

**Software:** Balkeshwar Singh.

**Supervision:** Gyanendra Kumar Singh, Balkeshwar Singh.

**Validation:** Balkeshwar Singh.

**Writing – original draft:** Manoj Kumar.

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
