## [Decision Letter · Decision Letter 0]

Dear Dr. Singh,

Thank you for submitting your manuscript to PLOS ONE. After careful consideration, we feel that it has merit but does not fully meet PLOS ONE’s publication criteria as it currently stands. Therefore, we invite you to submit a revised version of the manuscript that addresses the points raised during the review process.

We look forward to receiving your revised manuscript.

Kind regards,

Kelong Fan

Academic Editor

PLOS ONE

Journal Requirements:

Reviewers' comments:

Reviewer's Responses to Questions

**Comments to the Author**

1. Is the manuscript technically sound, and do the data support the conclusions?

Reviewer #1: Partly

Reviewer #2: Partly

Reviewer #3: Yes

2. Has the statistical analysis been performed appropriately and rigorously?

Reviewer #1: Yes

Reviewer #2: N/A

Reviewer #3: Yes

3. Have the authors made all data underlying the findings in their manuscript fully available?

Reviewer #1: No

Reviewer #2: Yes

Reviewer #3: Yes

4. Is the manuscript presented in an intelligible fashion and written in standard English?

Reviewer #1: Yes

Reviewer #2: Yes

Reviewer #3: No

Reviewer #1: Comments:

The manuscript examines the optical and electronic properties of SnOx/graphene oxide heterogeneous structures, with a focus on its sensitivity and selectivity for methane adsorption. A variety of research methods were used, several aspects of the manuscript could be improved to enhance clarity, impact, and rigor.

1.In the article, the SnOₓ/GO heterojunctions are classified into four types (Type-A to Type-D), but the specific numerical range or experimental determination method of oxygen content (x) is not clarified. It is only described vaguely as "high/low". Please use computational or experimental evidence to prove the specific numerical differences in the oxygen content of the four types of heterojunctions, please mark the chemical formula of SnOₓ or the range of the molar fraction of oxygen in the chart (as shown in Fig.1).

2. The phenomenon of increased reflectivity and decreased absorption caused by methane adsorption is only attributed to "electron interaction". Please use DFT to explain what factors the interaction sources of these phenomena are related to, such as charge density, etc.

3. The abscissa of Figure 6 is kcal/mol, and it does not mention what kind of energy it is. Adsorption energy? Binding energy? Or something else?

4. The main text mentioned the use of Monte Carlo calculation, but did not mention the Settings of relevant parameters such as temperature and pressure. Please provide the detailed parameters.

5. The calculation details of CASTEP are too few. The selection of pseudopotentials, whether van der Waals dispersion correction was chosen, and the sampling of K-points have not all been provided in the main text.

6. The unit cell parameters and vacuum layer of the four heterogeneous structures are not provided.

7. Please unify the use of picture descriptions in the main text and use them uniformly in accordance with the regulations of the journal, such as Fig. or Figure.

8. The resolution of Figure 1 and Figure 2 is too low, please give the color scale in the figures.

Reviewer #2: The article “Investigating SnOx/Graphene Oxide Heterostructure for Methane Sensing and Its Application as a Tunable Light Absorber for Optoelectronic Devices” investigates SnOₓ/graphene oxide (SnOₓ/GO) heterostructures for methane sensing and tunable light absorption, offering some novel insights into multifunctional material design. However, the manuscript requires significant revisions to address unclear method details, lack of experimental verification, insufficient support for some conclusions, and language expression issues, before it meets the standards for publication.

1. The synthesis method of SnOx/GO composites is not mentioned in this work. At least, referencing established preparation approaches from prior studies should be included. Additionally, the material stability under prolonged methane exposureand ambient storage remains unvalidated.

2. The conclusions of this study are solely based on computational simulations, lacking essential experimental validation. For instance, the actual methane-sensing performance of SnOx/GO heterostructure has not been experimentally evaluated. We recommend supplementing experimental verification or citing relevant experimental studies on analogous systems to evaluate the reliability of the computational findings.

3. Although computational simulations validate that the SnOx/GO heterojunction enables ppm-level trace methane detection, it fails to specify the exact limit of detection (LOD) or provide experimental validation. Furthermore, a comparative analysis with mainstream methane sensors in terms of sensitivity, or LOD is notably absent.

4. The selectivity validation is absent, as the study exclusively examines methane response without addressing possible interference from co-existing environmental gases (e.g., CO2 and H2O). Neither experimental evaluations nor computational analyses are provided to confirm the material's specificity under practical operating conditions.

5. The study highlights the application value of this research for photoelectrochemical systems and device engineering, yet fails to specify practical implementation strategies for SnOx/GO heterojunctions in such applications.

6. The manuscript contains redundant citations, with References [19], [20], and [23] all referring to the same publication. We recommend delete these duplicate references to comply with standardized citation practices.

7. Some phrasing issues in the manuscript should be further improved before publication.

Reviewer #3: This manuscript systematically investigates the optoelectronic properties and methane sensing mechanism of SnOx/GO heterostructures. By tuning oxygen content and methane adsorption, it reveals a synergistic enhancement in material performance, offering new insights for the design of broadband photodetectors. However, the theoretical model lacks validation of key parameters, such as the uniformity of oxygen distribution and the diversity of adsorption configurations, and it does not incorporate experimental data to support the simulation results, raising concerns about the feasibility of performance optimization. It is recommended to supplement the study with interfacial stress analysis of the heterojunction, cross-sensitivity tests with multiple gases, and research on scalable fabrication processes to enhance the engineering applicability of the findings. Here are the detailed comments.

1. The energy distribution analysis shows that the system stabilizes at three discrete energy states of -8.3, -8.8, and -9.4 kcal/mol. However, how do these energy states specifically influence the photoabsorption or methane sensing performance of the SnOx/GO system? For instance, does a lower energy value (such as -9.4 kcal/mol) indicate stronger adsorption stability? Do the different energy states correspond to particular surface configurations or charge transfer mechanisms?

2. The paper emphasizes the high sensitivity of the Type-II heterostructure in methane sensing (e.g., extinction coefficient up to 8.0), but lacks experimental validation data such as response time, selectivity, and repeatability. For example, is this material affected by humidity or interference from other gases in real environments? Are there practical sensor device tests (e.g., detection limit, linear range)? Is there a gap between theoretical simulation results and actual sensing performance?

3. The conclusion states that the SnOx/GO heterostructure holds potential in optoelectronics and gas sensing, but does not discuss the limitations for practical applications. For instance, does the high absorption coefficient of Type-C also entail issues of thermal stability or durability? Are there process challenges in scaling up the fabrication of SnOx/GO heterostructures? Moreover, has the long-term stability or environmental adaptability of methane sensing been evaluated?

4. The heterostructures are categorized into four types (Type-A to Type-D) based on the oxygen molar fraction of SnOx. Why is oxygen content chosen as the classification criterion? How does this classification influence the differences in optoelectronic properties of the materials?

5. Experimental results show that the Type-C heterostructure exhibits the highest absorption coefficient (~1.8×10⁵ cm⁻¹) near 100 nm, while Type-II shows the highest extinction coefficient (~8.0) near 1000 nm. Can these properties be synergistically optimized by tuning the oxygen content or methane adsorption level? Is it possible to balance both characteristics?

6. The study notes that methane adsorption significantly alters the refractive index and extinction coefficient of the heterostructures, yet stability tests under actual gas environments are not addressed. In practical applications, would prolonged exposure to methane cause drift in the material’s optoelectronic properties due to chemical degradation or structural reconstruction?

7. Compared to pure SnO₂ or graphene-based sensors, what are the main advantages of the SnOx/GO heterostructure? The paper mentions "low operating temperature"—what is the specific value? Has its performance been experimentally compared with that of existing commercial sensors?

**Do you want your identity to be public for this peer review?** For information about this choice, including consent withdrawal, please see our Privacy Policy

Reviewer #1: No

Reviewer #2: No

Reviewer #3: No

---

## [Author Response · Author response to Decision Letter 1]

29 May 2025

Original Manuscript ID: PONE-D-25-15120

Original Article Title: “Investigating SnOx/Graphene Oxide Heterostructure for Methane Sensing and Its

Application as a Tunable Light Absorber for Optoelectronic Devices"

To: PLOS one , Editor

Re: Response to reviewers

Dear Editor,

Thank you for allowing a resubmission of our manuscript, with an opportunity to address the reviewers’

comments.

We are uploading (a) our point-by-point response to the comments (below) (response to reviewers, under

“Author’s Response Files”), (b) an updated manuscript with yellow highlighting indicating changes (as

“Highlighted PDF”), and (c) a clean updated manuscript without highlights (“Main Manuscript”).

Best regards,

Manoj Kumar et al.

---

## [Decision Letter · Decision Letter 1]

Investigating SnOx/Graphene Oxide Heterostructure for Methane Sensing and Its Application as a Tunable Light Absorber for Optoelectronic Devices

PONE-D-25-15120R1

Dear Dr. Singh,

We’re pleased to inform you that your manuscript has been judged scientifically suitable for publication and will be formally accepted for publication once it meets all outstanding technical requirements.

Kind regards,

Kelong Fan

Academic Editor

PLOS ONE

Additional Editor Comments (optional):

Reviewers' comments:

Reviewer's Responses to Questions

**Comments to the Author**

Reviewer #1: All comments have been addressed

Reviewer #2: All comments have been addressed

Reviewer #3: All comments have been addressed

2. Is the manuscript technically sound, and do the data support the conclusions?

Reviewer #1: Yes

Reviewer #2: Yes

Reviewer #3: Yes

3. Has the statistical analysis been performed appropriately and rigorously?

Reviewer #1: Yes

Reviewer #2: Yes

Reviewer #3: Yes

4. Have the authors made all data underlying the findings in their manuscript fully available?

Reviewer #1: Yes

Reviewer #2: Yes

Reviewer #3: Yes

5. Is the manuscript presented in an intelligible fashion and written in standard English?

Reviewer #1: Yes

Reviewer #2: Yes

Reviewer #3: Yes

Reviewer #1: In this revised version, most comments of the referees have been well addressed, and the quality

of the manuscript has been substantially improved. Thus, I suggest the acceptance of this

contribution.

Reviewer #2: The authors have well addressed all my concerns, and the revised manuscriot can be considered publication in the journal.

Reviewer #3: Thank you for the opportunity to review the revised version of this manuscript. All the issues have been solved, and the manuscript is currently acceptable.

**Do you want your identity to be public for this peer review?** For information about this choice, including consent withdrawal, please see our Privacy Policy

Reviewer #1: No

Reviewer #2: No

Reviewer #3: No

---

## [Editor Report · Acceptance letter]

PONE-D-25-15120R1

PLOS ONE

Dear Dr. Singh,

I'm pleased to inform you that your manuscript has been deemed suitable for publication in PLOS ONE. Congratulations! Your manuscript is now being handed over to our production team.

Kind regards,

on behalf of

Dr. Kelong Fan

Academic Editor

PLOS ONE